# Oncogenic Role of HMGB1 as An Alarming in Robust Prediction of Immunotherapy Response in Colorectal Cancer

**DOI:** 10.3390/cancers14194875

**Published:** 2022-10-05

**Authors:** Huijiao Lu, Mengyi Zhu, Lin Qu, Hongwei Shao, Rongxin Zhang, Yan Li

**Affiliations:** Guangdong Provincial Key Laboratory for Biotechnology Drug Candidates, Institute of Basic Medical Sciences and Department of Biotechnology, School of Life Sciences and Biopharmaceutics, Guangdong Pharmaceutical University, Guangzhou 510006, China

**Keywords:** HMGB1, clinical prognosis, antitumor immune responses, cancer-associated fibroblasts (CAFs), tumor immune microenvironment (TME), immune checkpoints molecules (ICPs), methylated DNA, phosphorylated protein, colorectal cancer

## Abstract

**Simple Summary:**

The high mobility group box1 (HMGB1) protein operates as an alarm for danger. We explore the correlation between HMGB1 expression and the clinical survival status of patients receiving antitumor immunotherapy toward assessing the prognostic value in a multicancer context. We further conducted screening and identified four interacting genes that are targeted for binding by HMGB1-proteins and validate and contextualize the results in vitro. Additionally, HMGB1 function is also remarkably related to the tumor immune microenvironment and immune infiltration, especially regarding the high purity of stromal and immune cells in the tumor microenvironment (TME). In multicancer analysis, HMGB1 is also found linked with frequently genetic mutations and deletions, including in TMB and MSI which lead to resistance against antitumor immunotherapy and worse clinical prognosis among patients. The combination of HMGB1 expression with immune checkpoint molecules (ICPs), such as PD-L1, might present a novel immunology-based antitumor immunotherapy strategy.

**Abstract:**

Objective: To assess the correlation between HMGB1 expression and the patient prognosis in a multicancer context. Methods: The potential oncogenic role of HMGB1 was explored in forty tumors through the TCGA, GEO, and Oncomine datasets. We analyzed the clinical prognostic value and antitumor immunotherapy of HMGB1 in a multicancer context using GEO (GSE111636). Results: High expression of HMGB1 is present in multicancer cases, and its low expression is closely associated with the prognostic survival of patients, in terms of both overall and disease-free survival in ACC and LUAD. Further investigation revealed that the high expression of gastric and lung cancer is closely associated with low risk and better prognosis of patients based on COX and Kaplan–Meier analysis of OS, FP and PPS. HMGB1 expression was found to be significantly correlated with cancer-associated fibroblast and CD8^+^ T cell infiltration in the TME. The analysis of GO functional annotation/KEGG pathways indicates that HMGB1 may regulate tumor immunity-related pathways, such as the tumor immunotherapy response in colorectal cancer. The function of four genes as hubs are confirmed by in vitro HMGB1 knockdown which led to inhibition of cell proliferation and metastasis in SW620 and SW480 cells. Conclusion: HMGB1 is a potential novel biomarker for improving clinical prognosis and antitumor immunotherapy efficacy. CDK1, HMGB2, SSRP1, and H2AFV may serve as key nodes for HMGB1 in colorectal cancer.

## 1. Introduction

The high mobility group box (HMGB) family includes four members, HMGB1, 2, 3 and 4 with similar amino acid sequences and functional regions. This family was named for its high mobility in gel electrophoresis in 1973 [1]. However, each presents distinct expression patterns and participates in specific physiological and pathological cellular processes in diverse distributed tissues. Nuclear and extracellular HMGB2 mainly plays a critical role in DNA-bending and gene transcription, whereas nuclear HMGB3 only functions in the bending of target DNA and HMGB4 has only been found to function in developmental processes of the brain and pancreas [2,3]. HMGB1, a chromatin-binding nonhistone, is a highly conservative nucleoprotein [4]. The HMGB1 gene is localized on chromosome 13q12 and encodes a 215 amino-acids protein with a molecular mass of around 30,000; it is a highly conserved sequence and is formed of HMG-box1 and HMG-box2 [5,6]. It is known that excessive extracellular HMGB1 is released after cell death or through active secretion, and the subsequent formation of LPS-HMGB1 complexes induced by LPS activate TLR4 pathways to elicit inflammation and further activate the systemic immune response [7,8,9,10]. HMGB1, as a pro-inflammatory mediator, is a dual-function alarm protein used to notify the extracellular environment of cells in distress or danger, although it has also been shown to occasionally act as a suppressor of inflammation in certain tumors [11,12,13]. The regulation of HMGB1 function might affect and change the tumor biology and aid in the prevention and treatment of cancer in multicancer cases [14,15]. Recent publications indicate that high microsatellite instability (MSI) or deficiency is associated with worse prognosis during thermotherapy, and high levels of cancer neoantigens or checkpoint regulators are also exhibited [16,17].

The involvement of HMGB1 is complex in many cancers, e.g., nuclear or extracellular HMGB1 is engaged in tumor formation, progression, metastasis, and even in the response to chemotherapeutics [18,19,20]. Nuclear HMGB1 is always present in a fully reduced form in quiescent cells, enabling HMGB1 to be extracellularly converted into an effective activator of pro-inflammatory cytokine production by toll-like receptors-4 stimulation. In addition, the combination of HMGB1 and RAGE leads to the direct activation of NF-κB and subsequent cytokine formation, resulting in cell pyro-apoptosis and an inflammatory response and subsequent overactivation of the immune response [21,22,23]. HMGB1 is a disulfide protein containing disulfide bonds with a REDOX function [4]. The REDOX status of the three cysteines in HMGB1 determines the subsequent biological activity. With the mechanistic complexity of tumorigenesis and progression, it would be highly meaningful to identify some valuable genes to perform a multicancer analysis and to discuss the relationship of both clinical prognosis and survival with potential biomarkers of antitumor immunotherapy.

It is known that immunotherapy, including ICPs, can stimulate the immune system, activate surface antigen recognition and removal of tumor cells by immune cells, so as to achieve the purpose of cancer treatment. Cytotoxic T lymphocyte-associated protein 4 (CTLA4) and programmed cell death ligand 1 PD-L1 (known as CD274) are immune checkpoint proteins expressed on tumor cells and tumor-infiltrating immune cells. As an immunosuppressive molecule, CTLA4 functions in the blockade of signal transduction and inhibits the activation of T cells. PD-L1 binds to PD1, which is expressed on numerous tumor-infiltrating lymphocytes (TILs) to inhibit T cell activation, thus enabling immune escape from tumor cells. Thus, PD-L1 may be an innate immunosuppressive negative modulator of the immune response. The response is significantly related to tumor progression due to the immunosuppression function prior to stimulating antitumor T cells, and TCRs of TILs are recognized by tumor surface antigens and then trigger PD1 expression, thus enabling the reactive expression of PD-L1 in tumor cells and specifically enhancing the potential of antitumor immunotherapy [24]. The TME frequently leads to PD1-mediated T cell exhaustion, which inhibits the antitumor cytotoxic T cell response [8]. Thus, anti-inflammatory molecules in the TME and the interactions between tumors and immune cells could interrupt the activation of immune cells and induce immune suppression, leading tumor cells to evade the immune check and attack, obviously facilitating tumor growth.

The tumor consists of stromal cells, tumor cells, and infiltrating immune cells, which interact and contribute to the development of an immunosuppressive microenvironment. These activated stromal cells show different patterns of expression for specific factors compared with stromal cells that are distant from the cancer. Abnormal stroma signaling in these tumors adjusts some hallmarks of the tumor, providing the energy and nutrition necessary to promote tumor growth, and they are involved in supporting tumor cell survival and expansion and thus contribute to tumor progression. Stromal cells change within the TME, including in terms of the input of inflammatory cells, alterations in the extracellular matrix (ECM) architecture, and promoting angiogenesis, cancer-enabling processes that stromal cells help to facilitate [25]. These infiltrating stromal and immune cells together constitute the tumor microenvironment.

The available databases of The Cancer Genome Atlas (TCGA) and accessible GEO project (https://www.ncbi.nlm.nih.gov/geo/ accessed on 2 May 2021) are sources of multifunctional and valuable genomics datasets [26,27]. As is well known, for various species, the datasets provide structure or gene-related functional analyses of HMGB1 based on a multi-perspective of physiology and clinical pathology. An in-depth study indicated that HMGB1 is an available multi-functional protein, including details of cellular biological characteristics and post-transcriptional modification functions, among others. Some bioinformatic analyses are conducted through the TCGA dataset, Timer2.0, the Oncomine database, and Kaplan–Meier Plotter, to name a few. Moreover, we also present some current cell- and animal-based experimental evidence for the correlation between HMGB1 and different types of cancer in Appendix A. However, evidence on the relationship between HMGB1 and various tumor types in multicancer cases based on big clinical data is still lacking. The results of the present study will contribute to a more comprehensive understanding of potential HMGB1-related molecular mechanisms underlying the clinical prognosis and possible pathogenesis of different cancer types. In this study, the survival and prognostic value of HMGB1-expression analysis, as well as clinical applications, for patients is demonstrated.

## 2. Materials and Methods

### 2.1. Data Acquisition and Analysis of High HMGB1 Expression in Tumors

The transcriptome profiling dataset and corresponding clinical data were gathered from the NCBI Gene Expression Omnibus (GEO) database (https://www.ncbi.nlm.nih.gov/geo/ accessed on 2 May 2021) using a dataset with the accession number GSE111636. The dataset was generated using GPL17586Affymetrix Human Transcriptome Array 2.0. The raw data were already processed; the normalized series matrix file was directly downloaded for further analysis. Moreover, to search for the gene expression of HMGB1 in different tumors and adjacent normal tissue via the TCGA dataset, we used the site http://timer.cistrome.org/ accessed on 15 May 2021 and the TIMER 2.0 web server. We also logged into GEPIA2 web http://gepia2.cancer-pku.cn/#analysis accessed on 20 May 2021 to obtain several cancers without normal or with highly limited normal tissues [28] in combination with Genotype-Tissue Expression (GTEx) by setting a *p*-value cutoff equal to 0.01 and the log2 fold change of with an absolute value equal to 1 [29]. We chose a violin plot to visualize HMGB1 in four different pathological statuses, including stage I, stage II, stage III and stage IV.

### 2.2. Various Survival Prognosis Analysis

We explored survival prognosis of HMGB1 in different tumors. We logged into the site of GEPIA2 web server for OS and DFS data by setting high and low Cutoff value of 50%. Log-rank *p*-value was calculated and less than 0.05 was significant statistically.

### 2.3. The Analysis of Genetic Alterations

We used cBioPortal via the site https://www.cbioportal.org/ accessed on 22 May 2021. [30,31] to query the “Cancer Type Summary” module and chose Cancer Study to obtain the HMGB1-related genetic alterations in different cancer types. According to the TCGA project, the website provided the HMGB1-3D dimensional structure, copy number alteration (CNA) analysis, and survival analysis. We continued to use the MUTATION function to explore HMGB1 mutation via searching “Comparison” functions. Moreover, we used the site https://kmplot.com/analysis/ accessed on 25 May 2021 to query the survival prognosis of UCEC via Kaplan–Meier plot analysis. Additionally, we also used the TIMER2.0 web server with the function of “Immune-Gene” to acquire the relationships among tumor-infiltrating immune cells and HMGB1 expression across various tumors. Moreover, algorithms including TIMER, CIBERSORT, CIBERSORT-ABS, QUANTISEQ, XCELL, MCPCOUNTER and EPIC were utilized for immune infiltration estimations by setting *p*-values and partial correlation [32] values using the Spearman method. The results are visualized as scatterplots and heat maps.

### 2.4. HMGB1-Related Gene Set Enrichment Analysis

We explored the STRING web server through the site https://string-db.org/ accessed on 26 May 2021 to search HMGB1-related genes by setting a condition of “Homo sapiens” as species for identified experiments, and we selected related gene numbers to >50. We then used the GEPIA2 web server to query the top 100 hub genes of HMGB1 by utilizing the “expression analysis” and “similar gene detection” modules. The *p*-value was less than 0.05, and the correlation coefficient (R) with Pearson correlation analysis was given. Moreover, we used an interactive Venn diagram web server [33,34] to search for the intersection relationship between HMGB1 binding and interacting genes. We then obtained integrated discovery through combination with KEGG by uploading all the gene-related lists to the DAVID dataset. The results are given as a chart of functions.

### 2.5. In Vitro Validation

The colorectal cancer (COAD) SW620 and SW480 cells were cultured in DMEM with ten percent fetal bovine serum form Bio, China. The siHMGB1 was transfected in 12-well culture plate; this lasted twenty-four hours. Total proteins of SW620 and SW480 cells were extracted and used for Western blot analysis. In order to assess the effect of HMGB1 on the metastatic and invasive ability of SW620 and SW480 cells, we also conducted gap closure assay. All experiments were repeated 3 times. Total RNA from the SW620 and SW480 cells was gained with the RNAiso™ Plus reagent (Takara Bio, Inc., Otsu, Japan). RT-qPCR was tested with 2X Color SYBR Green RT-qPCR Maser mix (Tsingke Biotechnology Co., Ltd, Beijing, China) via the given instructions. The relative mRNA level was calculated using the 2^−ΔΔC^^t^ method (Appendix A).

### 2.6. HMGB1 Gene Mapping and Protein Structure Analysis

First, we searched HMGB1 via UCSC-xena with the site http://genome.ucsc.edu/ accessed on 20 May 2021 genome browser through human Dec. 2013 (GRCh38/hg38) assembly [35]. We conducted an analysis of HMGB1 conserved functional domains in different species using the database of HomoloGene via https://www.ncbi.nlm.nih.gov/homologene/ accessed on 8 June 2021. Moreover, we acquired the phylogenetic tree of HMGB1 in some different species via https://www.ncbi.nlm.nih.gov/tools/cobalt/ accessed on 8 June 2021.

### 2.7. Gene Expression Analysis Based on the HPA

To explore the oncogenetic role of the HMGB1 level in multicancer types, we first used the dataset of the Human Protein Atlas (HPA), and the HMGB1 relative analysis under physiological conditions could be obtained. Moreover, to acquire the HMGB1 relative protein expression level in the plasma samples, we searched the mass spectrometry-based proteomics function of the HPA web server. In addition, we defined the “low specificity” as normalized expression (NX) for more than one in the type of region/tissue/cell but not upgraded in others. Further variations through the website (https://www.proteinatlas.org/ENSG00000189403-HMGB1 accessed on 10 June 2021) were attempted.

### 2.8. Oncomine Database Show the HMGB1 Gene Expression Analysis

Searching https://www.oncomine.org/resource/login.html accessed on 15 June 2021, We first entered into online dataset of Oncomine to obtain the data of HMGB1 gene expression in different tumors and normal tissues by setting the threshold of *p*-value *<* 0.05, median rank, fold change is equal to 1.5. All of the pooling analyses were given in at least ten comparisons.

### 2.9. Analysis of Prediction of HMGB1 Protein Phosphorylation Sites and TMB/MSI

The online dataset of PhosphoNET (http://www.phosphonet.ca/default.aspx accessed on 23 June 2021) was used to acquire the predicted phosphorylation features of the S35, S39, S42, Y78, S100, Y109, S121, and Y162 sites via the protein name “HMGB1”. We then explored the role of TMB/MSI via the website of (http://sangerbox.com/Tool accessed on 2 June 2021) and UCSC (https://xenabrowser.net/ accessed on 20 May 2021) in various cancers using the TCGA approach. A *p*-value (Spearman’s correlation test) less than 0.05 was considered to denote statistical significance.

### 2.10. Survival Prognosis Analysis of Kaplan Meier Plotter and DNA Methylation

Through http://kmplot.com/analysis/ accessed on 25 May 2021, we obtained a series of survival analyses of OS, PPS, RFS, DMFS and FP by setting the “auto select best-cutoff”, log-rank *p*-value and 95% confidence on the Kaplan–Meier Survival Plotter. We used the MEXPRESS tool via the site https://mexpress.be/ accessed on 25 June 2021 to analyze the methylation of HMGB1 at the DNA level based on the use of multiple probes (e.g., cg04198824, cg16037679, etc.) in different tumors via TCGA. The β value was obtained, and the Benjamini–Hochberg *p*-value was considered to indicate statistical significance. The cor (Pearson correlation coefficient) value was given. Additionally, promoter region probes were highlighted clearly to analyze the methylated status in CESC by searching with the MEXPRESS approach and boxplot to perform normalization of the chip data.

## 3. Results

### 3.1. The Flow Chart of Multicancer Expression Analysis of HMGB1

We aimed to analyze the potential conserved oncogenic role of HMGB1 in 33 types of cancer through the TCGA, GEO, and Oncomine databases. We found that HMGB1 may function in the regulation of cancer through cell cycle and DNA signaling pathway-related functions by GO/KEGG enrichment analysis. Then, STRING and VENN analysis indicated a key gene group for the model and further bioinformatics analysis indicated four genes as the hub genes in the module; subsequently, experiments in vitro were performed by the knockdown of HMGB1. Through in vitro cell scratching, RT-qPCR, and WB techniques, we also confirmed that HMGB1 knockdown led to inhibited proliferation and metastasis of SW620 cells and SW480 cells to be inhibited. For multicancer analysis of the conserved oncogenic role of HMGB1 in different tumors, our study was designed and analyzed according to the flow chart shown in Figure 1.

### 3.2. HMGB1 Shows High Expression in Different Cancer Types

In order to understand the influence of high and low HMGB1 levels in multicancer cases, we analyzed the different expression statuses of HMGB1 through the TIMER2.0 web server across various cancer types on the TCGA database, as shown in Figure 2a. The HMGB1 expression level in the tumor tissues was distinctly higher than that in the normal tissues including BLCA, CHOL, COAD, ESCA, GBM, HNSC, LIHC, LUAD, LUSC, PRAD, READ and STAD.

We first focused on the human HMGB1 functions as an oncogenic role (NM_012963 for mRNA or NP_002119 for protein, Appendix A). As shown in Appendix A, the HMGB1-protein structure is different among species such as *H. sapiens, P. troglodytes, M. mulatta,* etc. There are two main domains, namely the HMG_ box (cl00082) domain and HMG_box_2 (pfam09011) domain. The phylogenetic tree data show the correlated HMGB1 evolutionary relationship in Appendix A.

Then, we filtered for high expression of HMGB1 in DLBC, GBM, and LGG and THYM using the GTEx web server and setting the threshold *p* as less than 0.05 (Figure 2b). We also conducted HMGB1 expression analysis and obtained a close relationship in clear cell RCC and ovarian and colon cancer, as seen in Figure 2c. Meanwhile, a significant difference for other tumors was not observed (Appendix A). As shown in Appendix A, we tried to explore the HMGB1 level within the low RNA tissue specificity and low RNA cell type specificity by setting all consensus normalized expression values greater than one. Based on the HPA web server, GTEx, and FANTOM5, HMGB1 presented the highest expression in the bone marrow, followed by the tonsils, total PBMCs and granulocytes. The high expression of HMGB1 equal to 2.8 μg/L according to the HPA or Monaco or Schmiedel databases (Appendix A), indicates that intracellular HMGB1 protein may leak outward under physiological conditions. The results of pooling analysis show that HMGB1 displays higher expression in COAD (colorectal cancer) and BRCA (breast cancer) compared with normal tissues according to Oncomine, accessed via the site of www.ONCOMINE.org accessed on 15 June 2021 and visualized with analysis by multicancer types. (Appendix A–c).

According to the GEPIA2 web server, using the “Pathological Stage Plot” function, we obtained a close relationship between HMGB1 expression and the pathological stages of cancer with KIRC, SKCM, THCA, LIHC, and LUSC (Figure 2d and Appendix A).

### 3.3. Distinct Analysis of Clinical Survival with HMGB1 Expression

To obtain a deeper understanding of the levels of HMGB1 expression, we explored the potential relationship of HMGB1 expression with clinical prognosis in different multicancer patients in the databases of the TCGA and GEO projects (Figure 3a,b). The results show that HMGB1 is highly expressed with a worse OS (overall survival) in KIRC but better OS for ACC and LUAD cancers. Through disease-free survival analysis, we observed high HMGB1 expression with favorable prognosis for ACC, CESC, HNSC, LUAD and SARC. However, as shown in Appendix A, we noted a high HMGB1 level is associated with better prognosis in breast cancer, lung cancer, liver cancer, and gastric cancer but not ovarian cancer. High HMGB1 expression is linked with poor OS in BRCA with HER2-positive status, as seen in Appendix A. We also obtained high HMGB1 expression with better OS and PPS for breast cancer, gastric cancer, and lung cancer. However, we obtained the opposite result for ovarian cancer, where high HMGB1 expression is linked with poor prognosis of OS and PFS (Appendix A). We also performed relative clinical subgroup analyses (Appendix A). Taken together, these results show that significant differences in HMGB1 expression are linked with clinical prognosis in multicancer analysis.

### 3.4. HMGB1 Genetic Variation Affects the Survival Prognosis in Several Tumors

To further interrogate the genetic structure variation status of HMGB1 with various cancer types, we obtained the highest HMGB1 alteration frequency value of greater than 8% for DLBC, followed by GBM. We also obtained a copy number alteration frequency of more than 6% with colorectal cancer cases, and a copy number deletion of 2% frequency of HMGB1 with AML cases for the type, site and case number of the HMGB1 genetic structure variation, as shown in Figure 4a,b.

We also obtained the missense mutation of HMGB1 with R163*/Q site mutation in the HMG_ box from R (arginine) to Q (glutamine) at 163 sites of the HMGB1 protein. We detected one case of GBM and two cases of UCEC, as shown in Figure 4b, and the three-dimensional structure of HMGB1 is shown in Figure 4c. Moreover, we used TCGA multicancer (PANCAN, N = 10,535, G = 60,499), from which we obtained the ENSG00000189403 (HMGB1), and the samples derived from the Cancer Peripheral Blood function to explore the how genetic variation in HMGB1 is related with both clinical prognosis in different types of cancer and with microsatellite instability. The prognosis of UCEC showed a better OS with when HMGB1 was altered (Figure 4d). Additionally, we explored the association between HMGB1 gene mutation and TMB or MSI (Appendix A) in considering a quantifiable biomarker based on gene mutations in various types of cancer, in which we examined several tumors in the TCGA dataset [36,37,38,39]. The results show that TMB is negatively correlated with HMGB1 alteration for THYM and THCA but there is a positive correlation for STAD, UCEC, BLCA, OV, PRAD and PCPG. We also obtained positive results of MSI for READ, THCA, STAD, UCEC and HNSC as well as ESCA, but the opposite was found for LUSC, LUAD and GBM. These results deserve further study.

### 3.5. Negative Relation between DNA Methylation and Gene Expression of HMGB1 in CESC

We first obtained the correlation of HMGB1 DNA methylation with pathogenesis in various cancer types via the TCGA project. We used the MEXPRESS web server and obtained a close, negative association of HMGB1 methylated DNA with probes corresponding to the non-promoter region, such as cg11047295, cg00589914, and cg25319276. We then continued to analyze the HMGB1-related chip file to normalize the chip data of adjacent normal cervical cancer (*n* = 24) and CESC tissues (*n* = 171) and visualized the results as a violin plot, as seen in Appendix A.

### 3.6. HMGB1 Protein Phosphorylation Sites and Different Expression Levels in Several Tumors

The levels of HMGB1 phosphorylation in normal and primary tumor tissues were analyzed for BRCA, clear cell RCC, LUAD, and UCEC via the CPTAC web server. Figure 3c presents a summary of HMGB1 phosphorylation sites. S35 was localized in HMG1_ box with cancers of BRCA, UCEC, and LUAD, followed by a considerably increased phosphorylation level of the S100 locus within the HMG2_ box domain for BRCA but not LUAD and ccRCC [40]. S100 of HMGB1 affects intracellular functions, including cell cycle progression and cell growth, among others [41,42]. In addition, S100 may serve as an inflammation marker of disease activity [43,44]. Through http://www.phosphonet.ca/default.aspx accessed on 23 June 2021, we analyzed the CPTAC-identified phosphorylation of HMGB1 and found that HMGB1 phosphorylation was experimentally correlated with previous presumptions (Appendix A) [32]. This finding could be explored in a further in-depth study of the potential role of S100 phosphorylation and molecular analysis in tumorigenesis.

### 3.7. HMGB1 Expression Was Differently Correlated to Tumor-Infiltrated Immune Microenvironment in Multicancer Types

Tumor-infiltrating immune cells perform a vital role in the initiation, progression or metastasis of some cancer types [45,46]. Cancer-associated fibroblasts (CAFs), mainly producing the interstitial matrix, function in different types of stromal cells in the tumor microenvironment [46,47]. Furthermore, according to some algorithms of TIMER, CIBERSORT, CIBERSORT-ABS, QUANTISEQ, XCELL, MCPCOUNTER, and EPIC, we explored the association of the immune-infiltrated level with HMGB1 in various tumors. We also found a positive association of immune-infiltrated CD8^+^ T cells with HMGB1 in LUAD, UVM, HNSC [HPV (Human papillomavirus)^+^], and THYM (Appendix A) through most algorithms [48,49]. Additionally, we found that the expression of HMGB1 is positively correlated with the infiltration of CAFs in HNSC, CESC, BRCA, KIRC, OV, and TGCT but negatively correlated in LUSC, STAD, and THYM (Figure 5) [50,51].

### 3.8. The Combinaton of HMGB1 Level and ICPs on Immunotherapy Efficacy in Human Multicancer Types

Stromal cells of the TME are vital components, and the percentage of stromal cells within the TME represents the stromal score. We first downloaded score data for three types of immune-infiltrating cells in forty tumors based on the TIMER web server and then chose the three most significant tumors. Figure 6a displays a remarkable association between HMGB1 expression and stromal score, including GBMLGG (Glioma), KIPAN, and KIRC. As shown in Figure 6b,c, high expression of HMGB1 has a significantly positive correlation with the immune-score as well as the estimate score (also known as tumor purity) in KIPAN, but a negative correlation in GBMLGG and LUSC.

Immune checkpoint molecules (ICPs) currently play a critical role in tumor immunotherapy. It is notable that immunosuppressive proteins such as CTLA4, PD-1, and PD-L1 inhibitory molecules, function to quell the reaction in the TME. Figure 6d displays the correlation between HMGB1 and ICP-gene coexpression in forty types of tumors. Expression of HMGB1 and PD-L1 is significantly positively correlated in UVM, OV, PAAD, KICH, KIPAN (Pan-kidney cohort (KICH, KIRC and KIRP)), KIRC, SKCM, UCEC, BLCA, BRCA, HNSC, LIHC, PCPG, PRAD, THCA, and LMAL and negatively in TGCT, CESC, ESCA, and NB. In addition, HMGB1 and CTLA4 expression are significantly positively correlated in UVM, OV, PAAD, KICH, KIPAN, KIRC, BLCA, BRCA, HNSC, LIHC, PRAD and THCA and negatively in TGCT, WT GBMLGG, LGG, UCS and THYM. Specifically, in WT, PD-L1 and CTLA4 yielded a contrasting result, and in-depth analysis is needed to formulate a rationale. As an immune checkpoint-activating molecule, HMGB1 was found to be significantly correlated with forty cancer types based on the TCGA database, positively activating the immune system to promote the inflammatory process and response to antitumor immunity [51]. We also evaluated the relationship between HMGB1 expression and immune checkpoint genes within the TME. Blockade of the immune checkpoints could boost the efficacy of immunotherapy and antitumor immunity [52,53,54].

At the same time, we investigated the immune function difference of HMGB1 and found that high expression of HMGB1 was remarkably related to the type-Ⅱ IFN response IFN γ and MHC class Ⅰ and CCR (Figure 6e). Additionally, we observed that the high expression of HMGB1 was significantly correlated with tumor dysfunction and exclusion (TIDE), and the results illustrate that the high expression of HMGB1 allows tumor cells to evade immune surveillance and promote tumor growth (Figure 6f). As shown in Figure 6g, we found that low expression of HMGB1 can effectively promote anti-PD1 immunotherapy.

### 3.9. Enrichment Analysis of HMGB1 Cell Signaling Pathways in GO and KEGG Databases

We first targeted HMGB1-binding correlated proteins with HMGB1 related genes as a set for KEGG analyses, searching for a potential molecular or functional mechanism of HMGB1 in tumorigenesis. To acquire the sum of 100 HMGB1-binding related proteins in Figure 7a, we present the related protein interaction network based on the STRING web server. We acquired the top 93 HMGB1-related genes from the GEPIA2 dataset. HMGB1 showed a positive association with CDK1 (R = 0.51), SSRP1 (R = 0.57), H2AFV (R = 0.53), and HMGB2 (R = 0.57) genes in several cancer types and these data were visualized as a heatmap (Figure 7c,d).

The KEGG data indicated that the “toll-like receptor pathway”, “spliceosome complex”, “damaged DNA binding” and “cell cycle” may exert vital effects in tumor pathogenesis and clinical prognosis through the KEGG and GO projects. Moreover, proteins, in their preference for binding to damaged DNA, are regulated by their DNA-binding domains. Additionally, we also explored the associations of HMGB1, CDK1, HMGB2, SSRP1, and H2AFV for COAD by immunohistochemistry (IHC) basedon the HPA database (Figure 7h).

### 3.10. HMGB1 Knockdown Inhibited COAD Cell Invasion and Migration In Vitro

The results of the gap closure assay, performed to investigate the effects of HMGB1 on the invasion and metastatic behaviors of SW620 and SW480 cells in vitro, indicate that HMGB1 knockdown decreases the proliferation and migration capabilities of cancer cells compared with control groups (Figure 8a). These same expression and distribution patterns for the above four genes in human COAD patients were verified by the RT-qPCR and Western blot results. The expression of HMGB2, H2AFV, CDK1 and SSRP1 was decreased in the HMGB1 knockdown group compared with SW620 in the control group (Figure 8b). We also attained similar results in the gap closure assay and RT-qPCR in SW480 cells (Appendix A). Moreover, HMGB1 knockdown led to significantly declined CDK1 expression in SW620 cells according to the Western blot (Figure 8c). All the above results indicate that the expression of HMGB1 is closely related to that of the four hub genes in colorectal cancer.

## 4. Discussions

HMGB1 is a nuclear protein with a highly conserved structure that is expressed in multicancer cases [55]. It operates as a multifunctional architectural protein with a series of intracellular and extracellular biological activities, including its binding receptors, subcellular functions, and post-translational modifications such as phosphorylation and acetylation [56]. On the one hand, HMGB1 modulates DNA damage repair as well as assists in the maintenance of genome stability through its function as a DNA chaperone in the nucleus. On the other hand, HMGB1 expression promotes autophagy, inhibits apoptosis, and modulates mitochondrial function in anticancer immune responses. Cells are frequently and constantly stimulated which may lead to injury or death. HMGB1, as an immunostimulatory agent, would increase innate immune cells and operate in the immune response particularly in inflammation due to aberrant signals. HMGB1 participates in the modulation of the immune response through the active secretion of immunocompetent cells, such as the TLR4 and NF-κB pathways. Some dead cells, such as necrotic and apoptotic cells, would release an amount of extracellular HMGB1 to combine with some pro-inflammatory molecules such as lipopolysaccharide to participate in the modulation of the immune response. Overall, the results from previous publications are consistent with the analysis of the KEGG pathways and GO functional enrichment of HMGB1-associated targeting genes. HMGB1 positively modulates multicancer-related pathways, e.g., extracellular HMGB1 increases the production of cytokines such as interleukin and interferon via NF-κ B, MAPK, and other pathways to promote cancer cells’ proliferation and growth, DNA replication and nucleotide excision repair.

Cytoplasmic HMGB1 boosts autophagy as well as suppresses apoptosis of cancer cells in anticancer immunotherapy. However, mutation or deletion of HMGB1 results in genome instability and initiates tumorigenesis. Notably, tumors were greatly related to microsatellite instability (MSI), tumor mutational burden (TMB) and copy number variations. Additionally, the genetic mutation or deletion of HMGB1 might suppress autophagy or promote apoptosis as well as tumorigenesis. However, disulfide HMGB1 plays a vital role in the modulation of the innate immune response via a special HMGB1 REDOX isoform. The immune tolerance and immunosuppression of HMGB1 may be induced when extracellular HMGB1 is over-oxidized, leading to loss of the capacity to produce pro-inflammatory cytokines. Therefore, focusing on chromosomal architectural HMGB1 might provide a novel perspective in anticancer immunotherapy.

PD-L1 is induced by the TME, combined with PD1, which delivers negative regulatory signals to T cells, leads to T cells’ inability to recognize cancer cells, thus promoting immune escape. However, the deletion or mutation of HMGB1 leads to the low expression of MHC class I (MHC-I), which fails to present tumor-derived antigens to CD8^+^ T cells, resulting in the failure of cytotoxic T cells to effectively recognize and kill tumor cells. In addition, high expression of HGMB1 promotes the production and release of type II interferons (IFN-γ), which ultimately leads to promotion of the tumor immunotherapy response of PD-L1 in cancer cells. From the transcript datasets GSE111636, we examined the immunotherapy of HMGB1 against PD1 antibody drugs in patients with urothelial tumors, and we found that the effect of the high HMGB1 expression on the immunotherapy of urothelial tumors with PD1 antibody drugs was inefficient. Therefore, the combination of HMGB1 and ICPs might closely impact the immunotherapy efficacy in diverse cancers.

Certain published studies have shown a functional association between HMGB1 and clinically related diseases, especially different types of cancer. However, whether HMGB1 could function as a major multicancer biomarker in different tumors, and the pathogenesis of tumors on possible mechanisms, needs to be explored and supported with experimental evidence. Thus far, there are no publications reported on a multicancer analysis of HMGB1 in different cancer types. Therefore, our report here is the first such example, in which we conducted comprehensive multicancer analysis of HMGB1 in thirty-three tumors through the TCGA and GEO tools. HMGB1 showed high expression in tumors and the analysis of the clinical prognosis of HMGB1 presented several different results. We used the databases of the TCGA-LUSC and TCGA-LUAD projects and analyzed the association between high HMGB1 expression and OS and poor DFS in lung adenocarcinoma. Appendix A shows the high expression of HMGB1 with poor prognosis of OS, FP, and PPS, especially for LUAD. The dataset of the GEPIA2 project indicates a statistical association between highly expressed HMGB1 and poor OS. We found that high HMGB1 expression is associated with poor clinical survival in terms of OS, PFS and PPS in ovarian cancer (Appendix A). We also obtained an association between the high expression of HMGB1 and poor prognosis in terms of OS, DMFS and RFS, particularly focused on BRCA patients with HER2-positive status (Appendix A). Of course, further related evidence needs to be obtained to further confirm that the high expression of HMGB1 plays a vital role in some tumors or the results in resistance to tumor changes in normal tissue.

The research was performed via bioinformatics analysis and experiments, where HMGB2, H2AFV, CDK1, and SSRP1 were screened as hub genes. HMGB1 and HMGB2, the highly structurally similar family of HMGBs, operate in a series of cellular processes such as DNA repair and transcription. HMGB1 regulates tumor cell growth, migration, and proliferation and plays a role in different intracellular biological processes through various species, including spliceosomes and the cell cycle. For instance, HMGB1 promotes the invasion and migration of tumor cells for KIRC, and there is a significant relationship between HMGB1 expression and poor clinical prognosis. We also obtained a possible effect for the cell cycle and single- or double-stranded DNA binding in various tumors according to enrichment analysis. CDK1 could regulate normal cell cycle progression by promoting the transition from the G2 to the M phase. SSRP1 contains the well-characterized DNA-binding HMG-1 domain. H2A.Z is a strong H2A variant with two non-allelic genes including H2AFZ and H2AFV [57,58,59], which play a vital role in liver tumorigenesis by regulating key molecules in the cell cycle and EMT status [60]. From the abovementioned data analysis, these genes are found to be strongly associated with high HMGB1 expression in most tumors. Therefore, among all these genes, we conducted a related molecular functional experiment of HMGB1 in SW620 and SW480. The results are consistent with those of reported publications wherein HMGB1 knockdown reduced growth and migration in the colorectal cancer model [61,62].

However, there still exist limitations and shortcomings. In summary, more clinical and survival risk factors should be examined to support the results and allow the development of more precise and effective diagnosis and treatment.

## 5. Conclusion

In brief, our research group presents multicancer analysis of HMGB1 combined with the study of clinical survival prognosis, methylated DNA, phosphorylated proteins, tumor-infiltrating immune cells, TMB, and MSI in different tumors. We provide a relatively comprehensive introduction of HMGB1 as a hub gene and a potential therapeutic biological marker for cancer progression from the perspective of clinical tumor samples. In addition, CDK1, HMGB2, SSRP1, and H2AFV may serve as key nodes for HMGB1 in colorectal cancer.

## Figures and Tables

**Figure 1 cancers-14-04875-f001:**
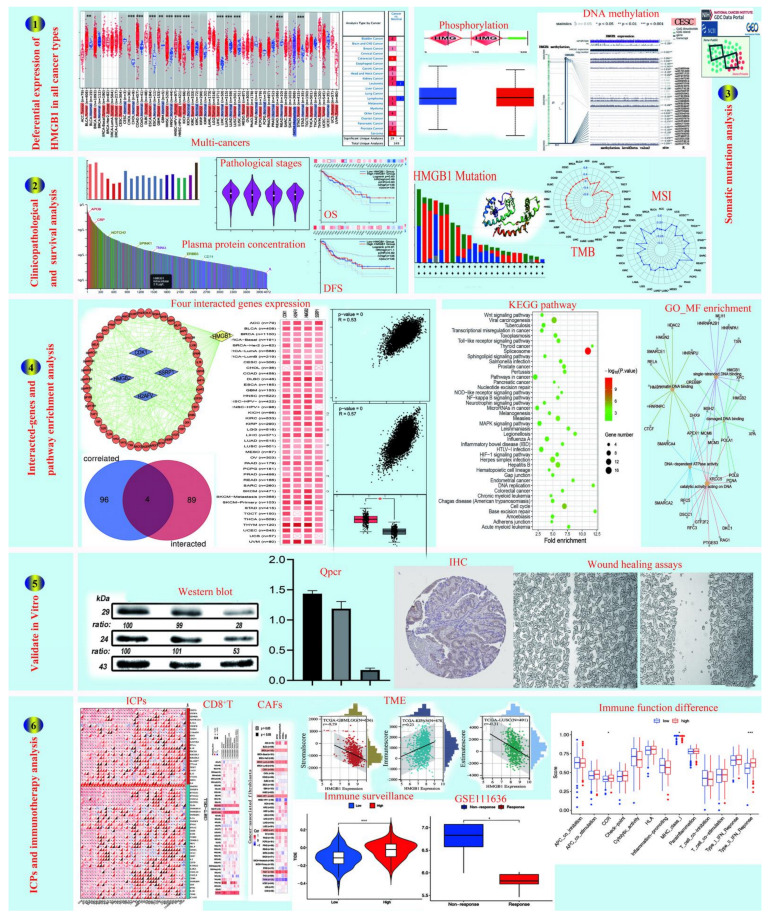
Flow chart of the HMGB1-related study design and analysis. (**1**) Differential expression of HMGB1 in all cancer types. (**2**) Clinicopathological and survival. (**3**) Somatic mutation analysis. (**4**) Interacted-genes and pathway enrichment analysis. (**5**) Validate in vitro. (**6**) ICPs and immunotherapy analysis. * Statistically significant *p*-value calculated (Pearson correlation analysis) *p <* 0.05, ** *p <* 0.01, *** *p <* 0.001.

**Figure 2 cancers-14-04875-f002:**
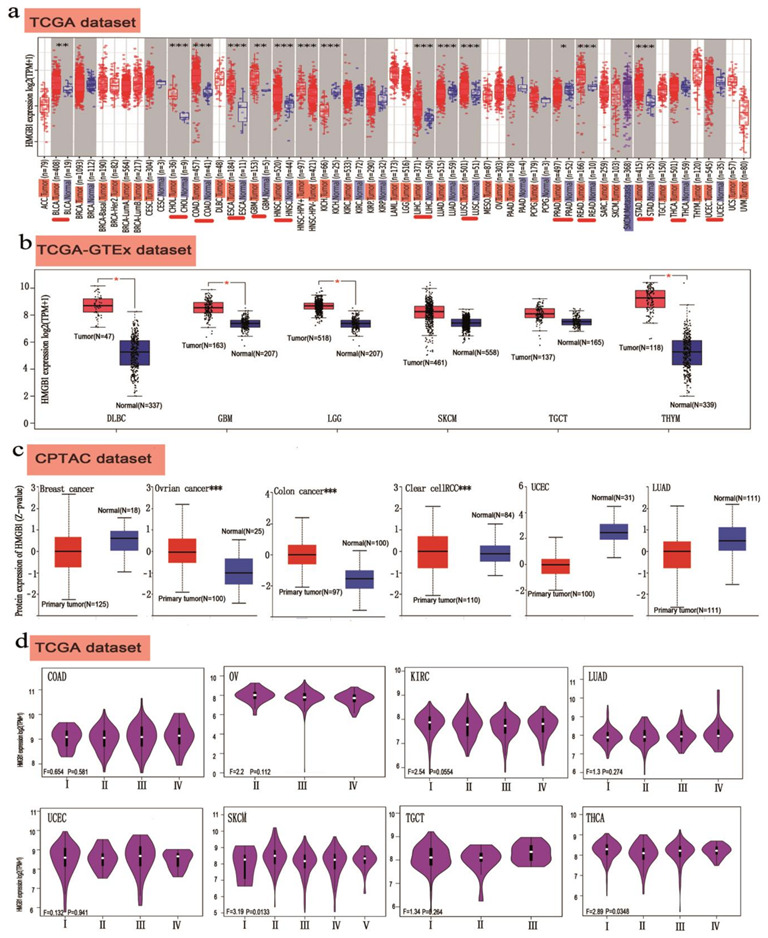
High expression levels of *HMGB1* gene in different cancers and at various pathological stages. (**a**) HMGB1 expression status in different cancers or specific cancer subtypes was explored, according to TIMER2.0 approach in the TCGA dataset. (**b**) Boxplot data of DLBC, GBM, LGG, SKCM, TGCT and THYM in GTEx. (**c**) Breast cancer, ovarian cancer, colorectal cancer, clear cell RCC, and UCEC in CPTAC. (**d**) HMGB1 expression levels were classified as stage I, stage II, stage III and stage IV for COAD, OV, KIRC, LUAD, UCEC, SKCM, TGCT and THCA according to the TCGA database. * *p <* 0.05; ** *p <* 0.01; *** *p <* 0.001.

**Figure 3 cancers-14-04875-f003:**
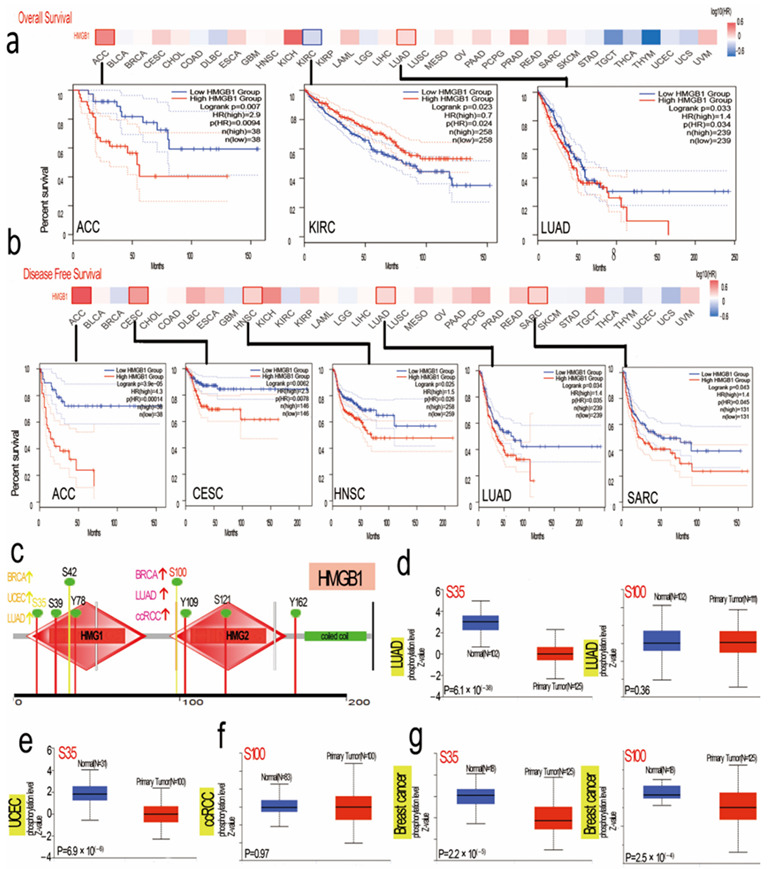
HMGB1-related gene expression and cancer survival prognosis and HMGB1-related protein phosphorylation analysis based on data from the TCGA database. (**a**) Overall survival with HMGB1 expression. (**b**) HMGB1-related disease-free survival. (**c**) HMGB1 phosphorylated protein (S15, T22, S35, S42, T51, S100, Y109, S121 and Y162 sites) in normal tissue and primary tissue. (**d**) LUAD (**e**) UCEC (**f**) clear cell RCC and (**g**) breast cancer.

**Figure 4 cancers-14-04875-f004:**
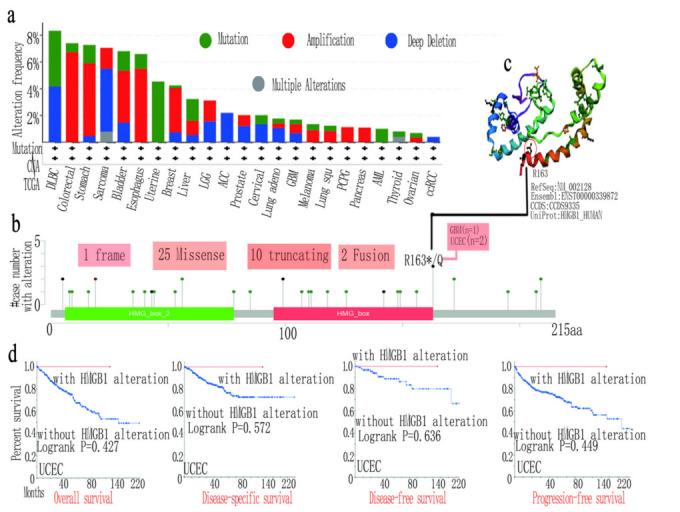
HMGB1-related mutation feature in various cancers in TCGA database. (**a**) HMGB1-related mutation type. (**b**) Multiple genetic mutation sites. (**c**) Highest alteration frequency (R163*/Q) in 3D structure of HMGB1. (**d**) Different survival statuses of UCEC with HMGB1 mutation.

**Figure 5 cancers-14-04875-f005:**
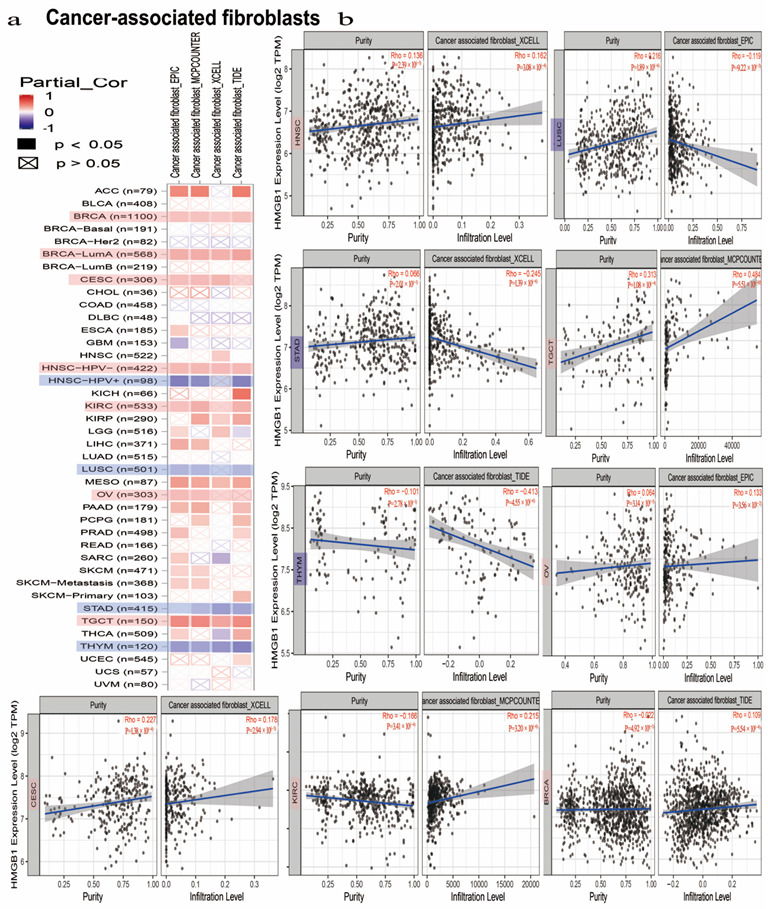
HMGB1 expression is closely linked with immune-infiltrated cancer-associated fibroblasts. (**a**) A heatmap of HMGB1 with immune-infiltrated level of cancer-associated fibroblasts in various cancers based on TCGA database. (**b**) A scatter plot of HMGB1 immune-infiltrated level of CAFs.

**Figure 6 cancers-14-04875-f006:**
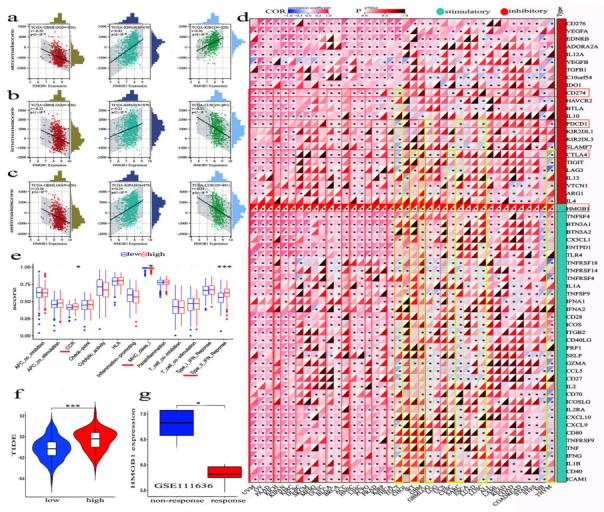
The correlation between HMGB1 expression and ICPs in multicancer types. (**a**) Stromal scores of TME (**b**) Immune scores of TME for tumor samples. (**c**) Estimate scores of tumor purity in multicancer cases. (**d**) All immune checkpoint-related genes were closely correlated with HMGB1 expression as evaluated by ssGSEA algorithm. The color indicates the cor value. * Statistically significant *p*-value calculated (Pearson correlation analysis) *p <* 0.05. (**e**) Analysis of immune function difference of HMGB1. (**f**) The correlation between expression of HMGB1 and tumor immune escape. (**g**) The correlation between the expression of HMGB1 and patients with therapeutic response to PD1 blockade immunotherapy. * Statistically significant *p*-value calculated (Pearson correlation analysis) *p <* 0.05, *** *p <* 0.001.

**Figure 7 cancers-14-04875-f007:**
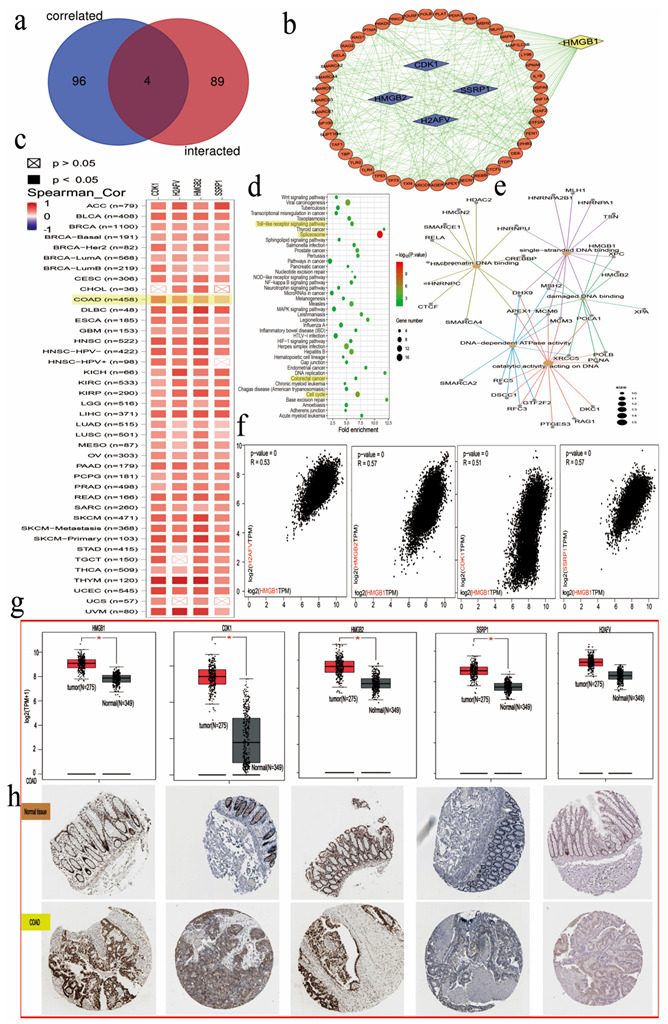
HMGB1-related gene enrichment analysis suggests four correlated genes. (**a**) A Venn diagram of the HMGB1-related selected genes. (**b**) Protein analysis on STRING tool. (**c**) A heatmap representing analysis of relationships between HMGB1 and four hub genes in multicancer types. (**d**) HMGB1-related KEGG pathway analysis. (**e**) The HMGB1-related centplot. (**f**) The corresponding scatter plots. (**g**) The expression of HMGB1-related hub genes (CDK1, HMGB2, SSRP1, H2AFV) in GEPIA database. (**h**) The changes in the expression of the 4 molecules were detected by immunohistochemical experiment in normal tissue and colorectal cancer (COAD) model. * Statistically significant *p*-value calculated (Pearson correlation analysis) *p <* 0.05.

**Figure 8 cancers-14-04875-f008:**
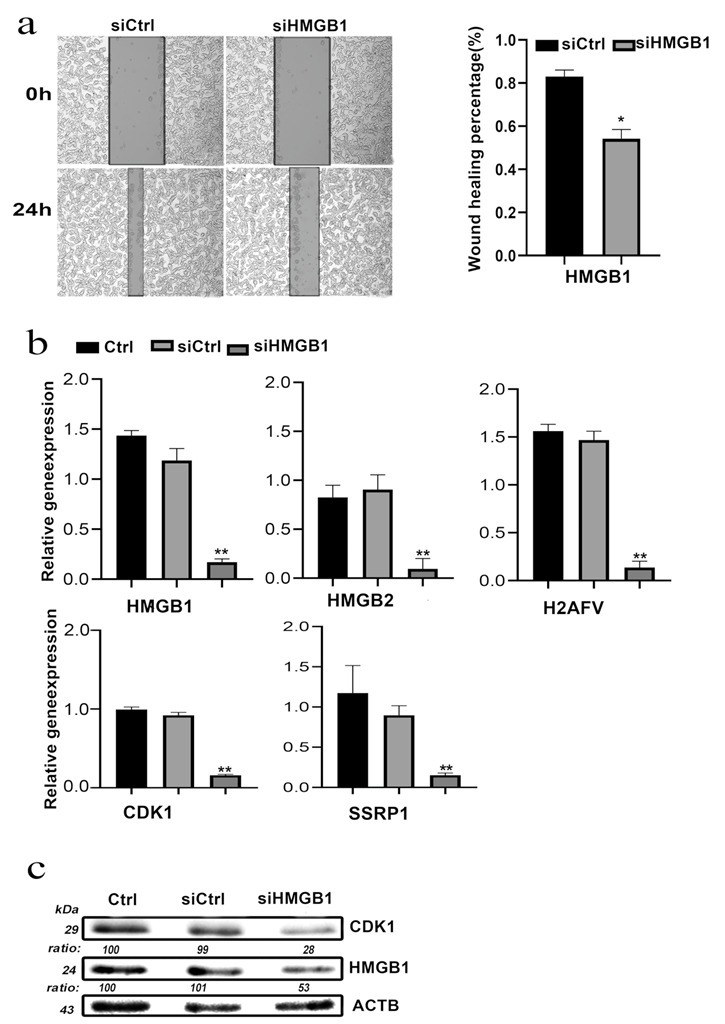
Functional validation of HMGB1 and expression of related genes in SW620 cell line. (**a**) Migration of SW620 cells assessed by wound healing assays. (**b**) HMGB1 knockdown efficiency. (**c**) HMGB1 knockdown reduces expression of its related gene CDK1 in SW620 cell line at protein level. * Statistically significant *p*-value calculated (Pearson correlation analysis) *p <* 0.05, ** *p <* 0.01.

## Data Availability

The authors certify that all the original data in this research could be obtained from public database. Other data used to support the findings of this study are included within the Appendix A. All the raw data of this study are available from the first author or corresponding author upon request.

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
