# Peer review of "Oncogenic Role of HMGB1 as An Alarming in Robust Prediction of Immunotherapy Response in Colorectal Cancer"

_cancers, 2022, doi:10.3390/cancers14194875_

Round 1
Reviewer 1 Report (Previous Reviewer 1)
Thank you for putting in so much effort and vastly improving this manuscript - I thoroughly enjoyed reading it and am pleased the review process has been so successful in this case.
Reviewer 2 Report (Previous Reviewer 3)
Thank you for making the suggested changes.
The manuscript has been sufficiently improved to warrant publication in Cancers.
This manuscript is a resubmission of an earlier submission. The following is a list of the peer review reports and author responses from that submission.
Round 1
Reviewer 1 Report
HMGB1 has been assessed as a potential oncogene across multiple cancer types and complex bioinformatic analysis performed to deconvolute immune cell types and ratios. The paper is extremely difficult to follow. The sequence of experiments makes sense however, data presentation could be clearer and explained much better throughout. For instance Figure 6 is very difficult to interpret what the data is showing. The choice of SW620 cells is unclear. To suggest an interaction with the immune system many more functional experiments in relevant models would be required. To the reader it is unclear where HMGB1 is expressed and how it regulates processes in cancer. I am convinced by the detailed analysis that there is a process worth exploring - however, the writing and presentation of this paper must change for publication. The authors having identified this potentially exciting pathway should look to determine mechanism of action in relevant models.
Reviewer 2 Report
The authors of this original article present a multicancer analysis of HMGB1 combined with clinical survival prognosis, methylated DNA and phosphorylated protein studies, tumor-infiltrated immune cell analysis, and TMB or MSI in different tumors. They made a relative comprehensive introduction of HMGB1 as the hub gene and a potential therapeutic biological target for the progression of colorectal cancer from the perspective of clinical tumor samples. They present a series of tests with a very complex approach, with high-level bioinformatics analyses, all validated on tumor tissue and cell lines.
The study is logically structured, well executed, and clearly presented. Their results are discussed with due moderation.
Minor language polishing is recommended.
I suggest accepting the article for publication.
Reviewer 3 Report
Authors in the present manuscript have reported the role HMGB1 as a potential therapeutic biological target for colorectal cancer. Correlation between HMGB1 and tumor-infiltrated immune microenvironment and immunotherapy responses are presented using online databases/platforms and algorithms in the study. The paper presents in vitro data suggesting that knock down of HMGB1 decreased migration and proliferation of colorectal cancer cells.
This is an interesting paper and needs major revision before publication.
Basic introduction of HMGB protein family could be included.
If possible, in vitro data for more than one colorectal cell line (other than SW620) or other cancers in which the protein is over expressed could be included.
Language editing is needed.